# Modification, Degradation and Evaluation of a Few Organic Coatings for Some Marine Applications

**Guang-Ling Song [1,2,3,\*]** and **Zhenliang Feng [1,†]**

1   Center for Marine Materials Corrosion and Protection, College of Materials, Xiamen University, Xiamen 361005, China; fengzhenliangzhenq@126.com
2   State Key Laboratory of Physical Chemistry of Solid Surfaces, College of Chemistry, Xiamen University, Xiamen 361005, China
3   School of Mechanical and Mining Engineering, The University of Queensland, Brisbane, QLD 4072, Australia
\*   Correspondence: glsong@xmu.edu.cn
†   The author finished the work in Xiamen University and currently affiliated with School of Materials Science and Engineering, China University of Petroleum (East China), Qingdao 266580, China.

**Abstract:** Organic coatings for marine applications must have great corrosion protection and antifouling performance. This review presents an overview of recent investigations into coating microstructure, corrosion protection performance, antifouling behavior, and evaluation methods, particularly the substrate effect and environmental influence on coating protectiveness, aiming to improve operational practice in the coating industry. The review indicates that the presence of defects in an organic coating is the root cause of the corrosion damage of the coating. The protection performance of a coating system can be enhanced by proper treatment of the substrate and physical modification of the coating. Environmental factors may synergistically accelerate the coating degradation. The long-term protection performance of a coating system is extremely difficult to predict without coating defect information. Non-fouling coating and self-repairing coatings may be promising antifouling approaches. Based on the review, some important research topics are suggested, such as the exploration of rapid evaluation methods, the development of long-term cost-effective antifouling coatings in real marine environments.

**Keywords:** organic coating; marine environment; corrosion protection; antifouling; coating evaluation; coating defects

---

## 1. Introduction

As an effective barrier to corrosion media, organic coating is one of the most popular corrosion protection approaches [1–8]. Since organic coating is usually more protective and relatively low cost, and can be more easily applied to large structures than other coating techniques, it is often the first option, particularly for heavy-duty anti-corrosion [9,10] in marine environments [11,12], where many factors, such as solar radiation, high salinity, hot–cold alternation, wet–dry cycling and micro and macro living species, can significantly exacerbate the environment corrosivity and dramatically accelerate the coating degradation [13,14]. Therefore, coatings used in such an aggressive environment must be extremely robust. For example, an organic coating for marine applications is usually a few hundred micrometers thick (typically around 250 μm), much thicker than a conversional coating [15]. Apart from the essential corrosion protection performance, the coatings for marine applications are usually required to be antifouling [16] or super-hydrophobic [17–19] to avoid the attachment of some creatures from sea.

Thus far, tremendous research efforts have been made to improve the corrosion resistance and antifouling performance for organic coatings in marine environments, including the design of new

polymer molecules or chains [20–22], change in coating constituents or additives [23–25], modification of coating micro-structures [26,27], decoration of coating surfaces [28,29], and adjustment of coating layers or composite interfaces [30–32], as well as the development of new techniques for coating characterization, evaluation and prediction [33–36]. Apart from these well-known achievements in the coating field, there are also some fragmentary but important progresses that may fundamentally influence the future research of marine organic coatings. This overview basically summarizes some of these key results obtained mainly in the past 15 years, aiming to illustrate the implications of the new findings and developments in possible marine applications. Since there have been many conferences [37–39], articles [40–42] and books [43–45] on the traditional coating systems for marine engineering, this review tries to look at some specific marine applications of organic coatings differently, aiming to establish a relationship between coating microstructure and marine application performance. Therefore, only a few fundamental issues that have not been reviewed before are addressed for some non-traditional marine coatings, rather than the practical behavior of traditional marine paintings or coating systems. From this perspective, it is expected that new and deepened understandings may be obtained for the behavior of organic coatings in marine environments.

## 2. Organic Coatings for Marine Applications

Generally speaking, a coating plays two basic roles in protecting its substrate [46,47]: (1) isolating the substrate from the environment and (2) changing the substrate surface to a different one with desired properties. The barrier effect in the first role is responsible for the corrosion resistance of organic coatings, while antifouling coatings are typically dependent on the surface functionality in the second role. These two roles can be combined in practice. For example, an organic coating with Zn powder pigment can serve as a barrier to water permeation to isolate a steel substrate and also act as a sacrificial anode to cathodically inhibit the corrosion of the substrate. When some antifouling additives are mixed in an organic coating, the coating will become anticorrosion and antifouling in the same time.

A marine coating system basically includes at least a pretreated substrate surface, a primer layer and a topcoat [48]. The substrate treatment is to remove contaminants and enhance the coating adhesion. The primer is the barrier against water permeation and the carrier of various species, such as anodic passivators, cathodic protectors, and/or inhibitors with particular functions. Traditionally, Zn epoxy-based and silica-based primers with Zn powder as the pigment have been widely used in marine environments. The topcoat, which is typically an alkyd, vinyl or epoxy film, is normally applied to protect the primer from UV irradiation or accident scratching or for a decoration purpose. To prevent the attachment or growth of a biofilm or some fouling, an antifouling coating may also be coated on the top. It should be noted that to date marine coatings are not simply limited on ships and platforms. Unprecedented and extreme applications, such as polar region navigation, deep sea exploration, navy military, etc., have brought many new materials, system designs, and function requirements into the coating industry. Therefore, the marine coating family is currently growing rapidly. For example, many advanced devices are now being more and more frequently used in marine atmospheric conditions [49,50]. They are usually powder coated or E coated very differently from a traditional marine coating. Therefore, the performance of these non-traditional marine coatings should also be included in the scope of contemporary marine coating research. Additionally, with more new metallic materials being increasingly used in marine environments [51,52], such as Ti alloys for submarines, Al alloys and Mg alloys for helicopters and fighters on warships and aircraft carriers, etc., the substrate in a coating system for marine applications is not limited to the steel anymore. Thus, the pretreatments of these new alloys, the organic coating materials and the corresponding coating processes, as well as the coating performance, will be quite different. These are also important topics.

The performance evaluation or service life prediction of a coating system is always one of the biggest challenges in the coating field [53–55], which can be seriously affected by the corrosion of the substrate surface, the permeability of the primer, and the disfunction of the topcoat. More specifically,

the defects formed during coating or introduced in service can facilitate the ingress of deteriorating and aggressive species into the primer coating, trigger the corrosion of the substrate and delaminate, and eventually rupture the coating, while the consumption of the antifouling species in the coating or the wearing of the special surface molecular microstructure of the coating can lead to the decay of antifouling performance. An organic coating can degrade or be damaged in some environmental conditions through various mechanisms. The deteriorating rate is critically determined by the microdefects for an anticorrosion coating, or the surface polymer molecular structure for an antifouling film. Therefore, in this review, the defects and polymer modification are focused.

## 3. Defects and Defect-Modification

Organic coatings inevitably contain macro- and microdefects. Under a certain condition, some of the defects are visible under SEM. For example, the SEM cross-section images of an epoxy E-coating and powder coating show that both the coatings are porous (see Figure 1). The E-coating contains more porous defects than the powder coating, and thus the latter is more corrosion resistant than the former on an Mg alloy [56].

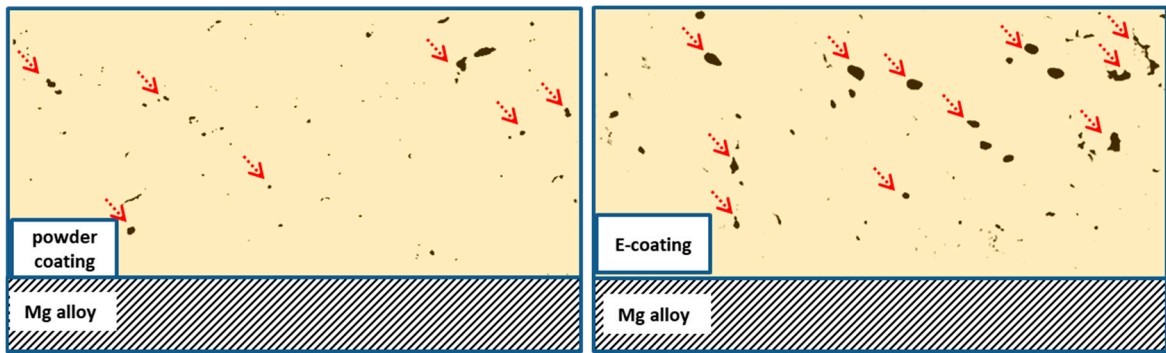

**Figure 1.** Schematic illustration of the cross-sections of epoxy powder coating and epoxy E-coating on a Mg alloy based on [56]. The defects may be across the entire coating, but present as pores or holes as pointed by arrows on the cross-sections.

It is normal that an organic coating can absorb some moisture [57]. A simple intake or ingress of environmental solution into the coating is usually a reversible process, which cannot damage the coating [58]. However, if the corrosive species, such water, oxygen and chloride, in the absorbed solution start to react with the substrate metal, then the accumulation of corrosion products on the substrate can eventually push off and distort the coating, resulting in coating blistering or spalling over the corroded areas, which will be an irreversible coating degradation and damage process. The irreversibly deteriorated coating defectiveness and eventually damaged coating will facilitate the ingress of corrosive species from the environment to more severely corrode the substrate. If the coating is originally defect free, then no aggressive media can reach the substrate to trigger the corrosion, and the coating damage will not take place. Hence, the size, number and distribution of the defects are critical to the protection performance of an organic coating.

Various macro- and microdefects may inevitably be formed during coating formation and introduced in service, which can actually act as short cuts for corrosive species to penetrate through the coating [59–65] and trigger the corrosion of the substrate. Obviously, the possibility of a defect continuously going through the entire coating can significantly reduce as the coating thickness increases, especially for a marine coating which is often a few hundred micrometers thick. However, this does not mean a thick organic coating is defect free, particularly after long-term service. It is possible to modify the defects through altering the coating curing conditions, such as temperature, pressure and airflow [66–68].

Apart from these natural environmental coating conditions well-known in industries, some artificial factors like electric, magnetic and electro-magnetic fields can also be employed to modify organic polymer coatings [69]. Feng et al. [70] controlled the airflow on the coating surface during curing to alter the defects in the coating to improve the coating anti-permeability. The airflow-modified thinner region of the coating was found to be more impermeable than the other thicker regions. The abnormal permeation behavior could be attributed to the different air pressures caused by the airflow (see Figure 2). The polymerization of the curing alkyd resin normally starts from the top surface layer [71]. A higher pressure can retard the solvent (or diluent) evaporation [72] from the surface and let the coating to fully polymerize, while the coating top layer will be cured more quickly or insufficiently polymerized at a low pressure. Moreover, faster evaporation at low pressure is likely to generate more cracks in the coating surface layer due to the poor polymerization [73]. Thus, the central region of the coating right facing the airflow with a higher pressure will have a lower density of defects, whereas the ring zone is more defective (see Figure 2). The reduction of coating permeability by airflow has not been realized in the coating industry. As alkyd is sometimes used as a topcoat in marine environments, this finding may be employed improve the coating protection performance in a selected area for some special applications.

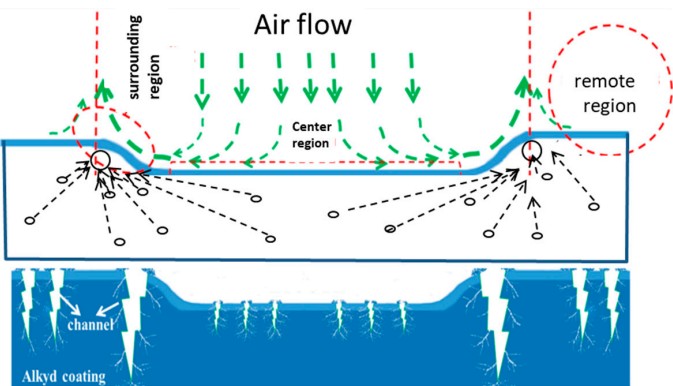

**Figure 2.** Schematic illustration of an airflow over an alkyd coating during curing, solvent evaporation distribution and possible micro-crack formation in the curing coating, adapted from [70].

Similarly, an electric field can also alter the microdefects locally in an organic coating, which is a traditional marine primer and topcoat, thus modifying the coating protection performance. A curing epoxy resin film under a local electric field can be attracted and repelled alternatingly, exhibiting an interesting oscillation behavior, and eventually forming a compression region [74]. A compression region was also found on an alkyd resin if it cured under a local electric field and the compression region had lower permeability in the cured coating (see Figure 3) [75]. However, such a local modification using an electric field may change the chemical compositions of a polymer coating [76] and may be too complicated to be used in the field [77,78].

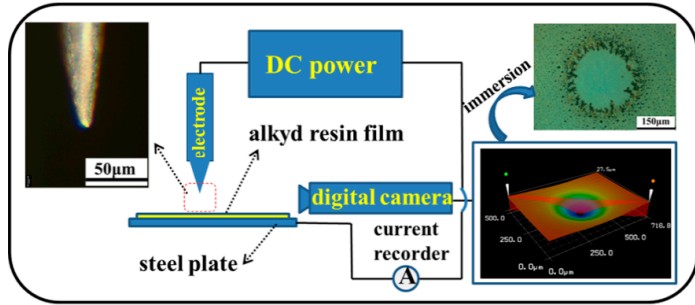

**Figure 3.** Schematic illustration of local modification of the surface morphology of a semiliquid alkyd film under a DC electric field [75].

## 4. Substrate Effect

Through the original defects formed during coating formation, the environmental media can reach the substrate after the coating is in service. Basically, an organic coating is bonded to the substrate physically and chemically through mechanical anchoring effect [79,80] and metal-organic/oxide-organic bonds [81]. Obviously, the corrosion of the substrate can significantly weaken its bond with the coating. If the substrate is corrosion resistant and the corrosion products are very slowly accumulated at the interface between the coating and substrate, then the coating degradation or failure, such as blistering and spalling, may be effectively delayed. In other words, the corrosion resistance of the substrate can also significantly influence the protection performance of a coating system, which has been verified in many studies [82–86]. It has been reported that a corrosion resistant substrate also has significantly improved corrosion resistance after coating [56,87].

In practice, organic coatings cannot protect an active metallic substrate as effectively as a passive one. For example, it is well known that Mg alloys are much more active than steel. Thus, coating delamination can usually be detected due to the severe corrosion of the substrate Mg alloy (see Figure 4) [56]. Therefore, the substrate pre-treatment is not only for better coating adhesion, it is also critical for the improvement of corrosion protection performance, particularly when the substrate itself is too active.

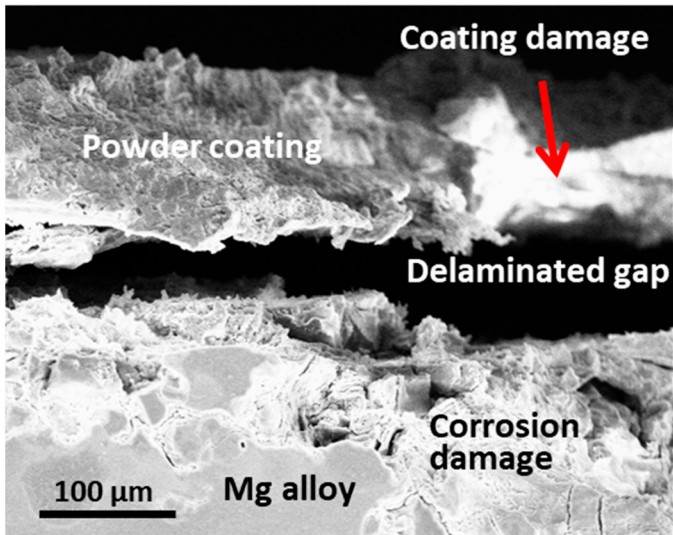

**Figure 4.** The substrate corrosion, coating damage and interface delamination for a coated AZ31 after 1000 h of salt spray, adapted from [56] (Copyright 2020, with permission from NACE International).

If the substrate has many anodic and cathodic sites that can form a large number of galvanic couples on the substrate surface, the protectiveness of the coating will be dramatically reduced at their joint region, because after corrosive solution penetrates through the coating, active galvanic corrosion will occur between the substrate couple. The corrosion damage of a coated joint is much more serious than that of a coated passive substrate [86,88,89]. It is an inappropriate idea in some industries to stop or prevent the serious galvanic corrosion around the joint of active metals simply by application of an organic coating in a conventional way.

The substrate corrosion may even occur while an organic coating is curing, which can adversely influence the protection performance of the coating after curing. For example, during polymerization of an epoxy coating, crosslink reaction occurs between the epoxy and curing agent [90–92]. The epoxy resin offers epoxy groups for the crosslinking [69], while the curing agent, including aliphatic, cycloaliphatic and aromatic amines or polyamides [93], provides active hydrogen to open the epoxy. In this period, environmental water vapor containing sufficient oxygen can be absorbed by the amines and amides in the curing agent. Therefore, when the mixture of the epoxy and amine is applied on the substrate metal,

the water dissolved in the curing agent will facilitate the electrochemical reaction of the substrate, resulting in corrosion damage of the substrate. Consequently, the corrosion products, including oxides/hydroxides of the substrate metal and hydrogen bubbles, formed on the substrate under the curing coating will decrease the coating adhesion, increase the coating porosity, and exacerbate the coating permeability [94]. It was found that after 1 and 4 days of salt spray (ASTM B117), no corrosion was visualized on an epoxy coated carbon steel (CS) or a magnetron-sputtered Mg (SM), whereas nearly half of the coating surface area was damaged on a magnetron-sputtered Mg layer covering over the carbon steel coupon (CS+SM) or a magnetron-sputtered iron layer covering over a magnetron-sputtered Mg layer (SM+SI) (see Figure 5). The magnetron-sputtered Mg layer covering over a carbon steel coupon (CS+SM) and the magnetron-sputtered iron layer covering over a magnetron-sputtered Mg layer (SM+SI) are much more active in corrosion than the magnetron-sputtered Mg layer (SM) and the carbon steel (CS) due to the micro-galvanic effect among the sputtered magnesium, iron, and steel coupon. They suffered from different degrees of corrosion damaged during epoxy coating curing, which significantly affected the protection performance of the cured epoxy coating exposed in salt spray.

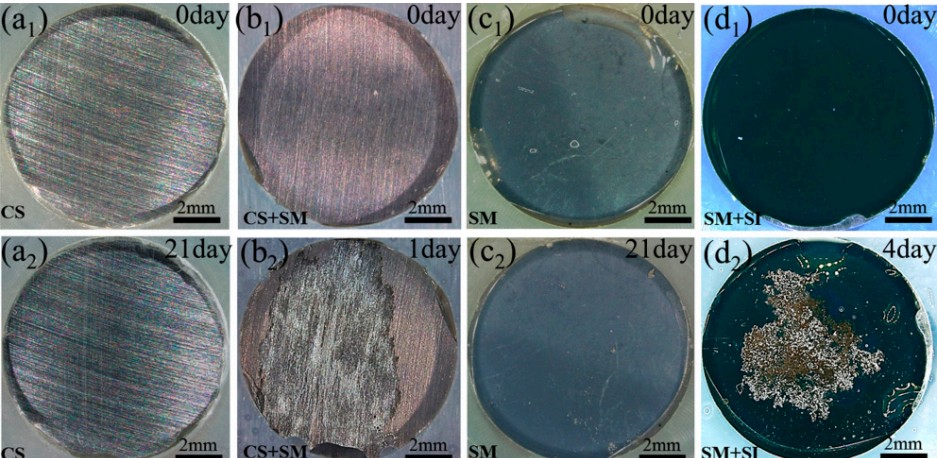

**Figure 5.** Corrosion damage of epoxy coated (**a₁,a₂**) carbon steel (CS), (**b₁,b₂**) magnetron-sputtered Mg layer (CS+SM) on carbon steel, (**c₁,c₂**) magnetron-sputtered Mg layer (SM) on glass and (**d₁,d₂**) magnetron-sputtered iron layer on magnetron sputtered Mg layer (SM+SI) after salt spray for different periods of time, reprinted from [94] (Copyright 2020 with permission from Elsevier).

The above results, including Figure 5, suggest that a thick epoxy primer in a marine coating system may adversely affect the substrate surface state during coating, particularly when the substrate is not a traditional marine steel. The primer must be stored with great cautions, and applied on the substrate and then cured strictly following the recommended practices at a suitable temperature and carefully controlled humidity to avoid early delamination. For a given substrate and an organic coating, a proper surface treatment before coating is therefore extremely important for the coating adhesion and protection performance [95,96]. In some cases, a simple heating or dehydration of the substrate before coating may even significantly promote the corrosion resistance of the coating [89]. Tremendous research and engineering efforts have been made to develop more cost-effective substrate pretreatments for coatings [97–99].

It should be noted even through the importance of the substrate surface state or the essentiality of the substrate pre-treatment has been well-known, the "pre-corrosion" of a substrate during coating polymerization [94] has not been realized in the coating industry. According to this new knowledge, a good surface preparation of the substrate before coating is not enough. It is necessary to take some additional measures to prevent the substrate corrosion when coatings are being cured.

## 5. Environmental Influence

When a cured organic coating is exposed in its service environment, it will degrade and eventually fail. If the environment is aggressive like marine, a coating system involving different coating layers, interfaces and the substrate metal will degrade rapidly and severely. Due to the presence of defects, the environmental factors can influence the organic coating degradation deeply and even the substrate corrosion.

### 5.1. Environmental Factors

Many marine factors can affect coating degradation and the substrate corrosion, such as the solar irradiation, salinity condensation, dry–wet alternation, and cold–hot cycling [100–103]. For example, the detrimental effect of heating and polymer crystallization on coating behaviors has been widely studied [104,105]. It is also well-known that ultraviolet (UV) irradiation can stimulate the change of polymer structure in organic coatings [106], and thus an organic coating normally degrades more rapidly under UV due to the photo-oxidation effect. UV facilitates the formation of macromolecular radicals and the subsequent reactions of the radicals with polymer macromolecules and oxygen, and the formed unstable oxygenated species can further evolve towards stable macromolecules containing oxygenated groups [107,108]. Therefore, the overall damage of an organic coating by UV irradiation is usually surface chalking and roughening [17]. The following sections simply review some factors whose influencing mechanisms on coating formation and protection performance have not been comprehensively understood.

### 5.2. Water

Apart from the well-known temperature, humidity is a critical factor that can substantially affect the protection performance of an organic coating [109,110]. Under immersion condition, coating swelling is a common behavior, which may result in narrowed cracks or pores in an organic coating. The water absorption induced coating swelling has been experimentally verified [111]. It is generally influenced by coating porosity and salt concentration [112–114]. Similar cases also include the addition of water repellence in some coating systems, the variation in osmotic pressure when a ship/boat navigation from a river to sea and vice versa, etc.

As mentioned earlier, the water from the environment can be dissolved in curing agent. This can enhance the electrical conductivity of the mixture the curing agent and epoxy resin during curing, electrochemically facilitating the micro-galvanic activity of the substrate. It has been reported [94] that an Mg–steel couple immersed in epoxy resin (A) curing agent (B) and their mixture ($A_3B$) has galvanic current densities increasing with increasing relative humidity (RH) due to the continuous absorption of water from the environment into the curing agent (see Figure 6). However, as the mixture ($A_3B$) of epoxy resin and curing agent polymerizes and water absorption reduces with time, the galvanic current density decreases (see Figure 6c). As the mixture cannot solidified thoroughly when RH is too high [115,116], the galvanic current stabilizes at 60% RH and 95% RH. Since the curing behavior influenced by the environment humidity can also actually affect the cured coating quality and thus the final coating permeability, the storage and application of curing agents in the coating industry should be handled carefully.

Generally, on a new structure the best practices must be followed, and the coating is not applied at high humidity. However, when the coating in service needs to be repaired in the splash, tidal and immersion zones, the humidity cannot be ideally controlled. The corrosion of the substrate during coating and curing affected by the moisture cannot be ignored.

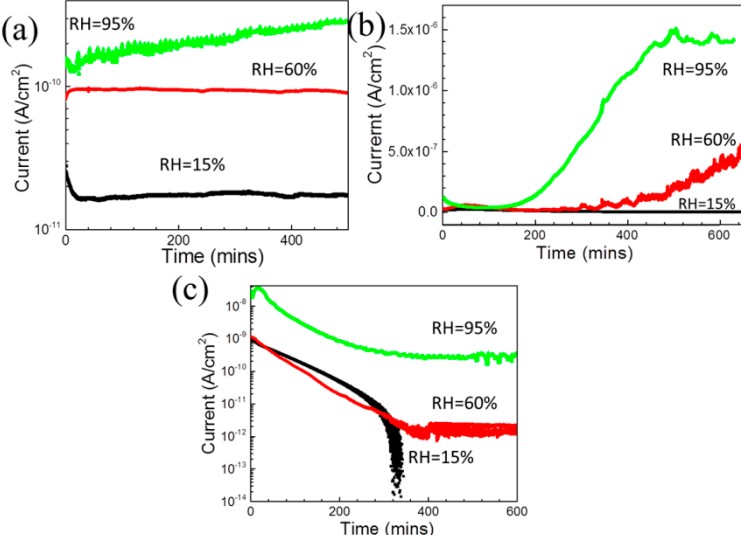

**Figure 6.** Galvanic corrosion current densities of coupled pure Mg and carbon steel immersed in (**a**) epoxy resin (A), (**b**) curing agent (B) and (**c**) their mixture (A$_3$B) at different relatively humidity levels at 20 °C, reprinted from [94] (Copyright 2020, with permission from Elsevier).

### 5.3. Salt

NaCl is one of the major components in seawater. Precipitated salt crystal particles may grow inside the pores or defects of a coating during drying in a wet–dry cycle, similar to the salt crystallization in a porous material [117–119]. This may lead to salt crystallization-assisted degradation of the organic coating in marine environments, which had not been realized before until the investigation by Feng et al. [120]. When fast evaporation of the remaining solution on a coating surface particularly in a windy day leads to precipitated NaCl crystal particles growing in the microdefects or pores in the surface layer of a coating, a force ($\triangle$p) would be built up after the NaCl crystal particles in contact with the pore inner walls [117,121,122]. This pressure will enlarge the defects or pores, deform the coating surface layer, generate cracks and eventually damage the coating [123], resulting in a significantly roughened coating surface.

The finding of salt crystallization induced coating damage may open up a new topic in organic coating research. Some inhibitors that can combine with chloride or sodium to retard the salt-precipitation may be applied on the coating surface to mitigate the coating damage.

### 5.4. Synergistic Effect

Synergistic effect is known as a combined effect of different environmental factors stronger than simple addition of all the individual influences. In marine environments, the photo-oxidation resulting from solar irradiation may trigger chain scission of polymer molecules [108,124,125]. The wet–dry cycling can exert a swelling–shrinking effect on the coating, consequently degrading or even breaking down the coating [126,127]. Meanwhile, the hydrolysis of organic molecules may occur to facilitate the chain scission in the organic coating. When an epoxy resin surface is subjected to alternate UV irradiation and salt water immersion ($E_{NaCl+UVA}$), the polymer molecules with broken cross-linking bonds by UV irritation can be more easily hydrolyzed, and the hydrolyzed polymer molecules can be more easily dissolved for UV to oxidize the new exposed surface. In this case, the alternate salt water immersion and UV irradiation ($E_{NaCl+UVA}$) will more seriously accelerate the degradation of the epoxy resin than the sum of continuous salt water immersion effect ($E_{NaCl}$) and continuous UV irradiation effect ($E_{UVA}$). Experimentally, the epoxy resin surface after the alternate water immersion and UVA irradiation was indeed much rougher (see Figure 7). There was an obvious synergistic effect between the UV irradiation and the NaCl solution immersion on the coating degradation [120].

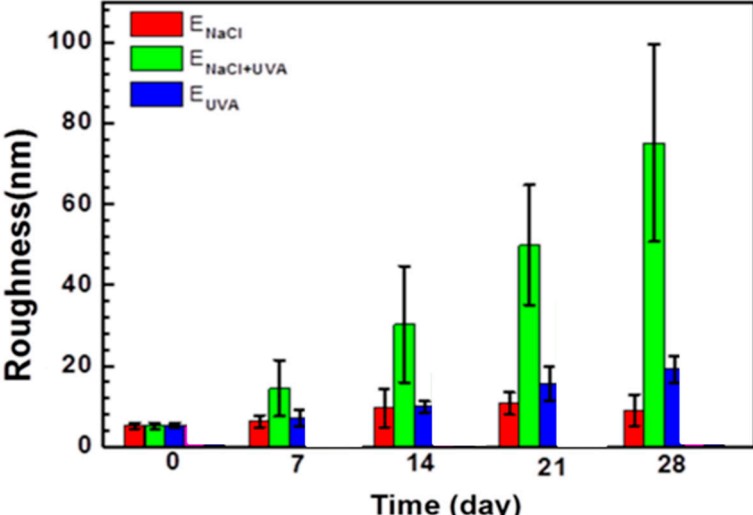

**Figure 7.** Surface roughness of epoxy resin after continuous salt water immersion ($E_{NaCl}$), continuous UV irradiation effect ($E_{UVA}$) and alternate UV irradiation and salt water immersion, reprinted from [120] (Copyright 2020 with permission from Elsevier).

Based on the understanding of the synergistically accelerated effect of salt water immersion and UV irradiation, the degradation of an organic coating in the marine tidal and splash zones may be better predicted. For example, it has been estimated that the acceleration effect of the $E_{NaCl+UVA}$ on epoxy degradation in the lab is approximately 20 times stronger than the natural environment in Xiamen island if the other factors can be neglected [120]. In other words, a one-month lab test will be equivalent to a 20-month island exposure. Thus, with the understood acceleration effect and obtained short-term lab test results, the long-term coating degradation behavior in the field may be predicted.

## 6. Evaluation

Many techniques have been employed to evaluate the protection performance of organic coatings, among which, electrochemical methods are usually more attractive to researchers in the lab, because they are relatively more rapid than the others based on the corrosion reaction of the substrate metal, as well as the relevant processes through the coatings. Currently, in the lab, the most widely used technique is AC electrochemical impedance spectroscopy (EIS) [128–130] and electrochemical noise (EN) [131,132]. The former can measure the coating resistance and substrate corrosion resistance, while the latter basically provides some local damage activities of the coating. Technically, the former is more reliable than the latter in terms of the resistance against environmental noise, and hence can be used in the field and have been attracting more attention from coating researchers and engineers. As these electrochemical techniques measure instantaneous changes in the substrate and coating, they are good to be employed to monitor the degradation and damage of a coating system [133–135]. There are numerous non-electrochemical coating evaluation methods. Instead of going over all the well-known traditional approaches, this review simply list a few methods that are scientifically interesting in coating research. They include some surface, microstructure, composition, and chemical state analyzing techniques [136–138], such as the Fourier Transform Infrared Spectroscopy (FTIR) [139] and the Confocal Raman Microscopy (CRM) [140] to estimate the bonds and funcational groups involved in the coating as the polymer molecular changes, the Atomic Force Microscope (AFM) [141,142] to detect the coating surface structure and state. These techniques are more focused on the organic coating than the substrate. They are basically used only in the lab, not for field test. Either the electrochemical or the non-electrochemical methods are aiming to rapidly evaluate the corrosion resistance for an organic coating system in service or even predict its service life [143]. However, due to the lack of information on

the coating defect size, number and distribution, the evaluation and prediction of coating performance are actually very difficult.

## 6.1. Simulation and Rapid Evaluation

There are already many conventional lab simulation and evaluation techniques and standards developed for organic coatings [144,145]. Some have been widely used in the industry, while some are used in the lab only for a fundamental research purpose. In this section, a few most popular and recently developed ones are reviewed.

### 6.1.1. Salt Spray

Salt spray is the most commonly used in the lab to simulate the marine atmospheric, splashing and tidal zones [146–148]. However, it cannot speed up the coating degradation very efficiently. It may still be used on some organic coating systems with a very active metal as the substrate, as the corrosion damage of these systems may occur quickly in salt spray [149]. For example, the improved anticorrosion property of a polyurethane coating containing $Cr_2O_3$ nanoparticles on carbon steel was successfully evaluated by 624 h of salt spray [150]. The standard salt spray has also be used to evaluate four differently coated Mg alloys. After 1000 h of the test, the epoxy powder coated AZ31 and AZ61 showed higher corrosion resistance but worse adhesion than that of the epoxy E-coated AZ31, while the epoxy E-coated + epoxy powder-coated AZ31 had the best corrosion protection performance [56,151]. How to speed up the degradation process of a coating system in the salt spray condition is currently an interesting topic in coating research and engineering.

### 6.1.2. Immersion and EIS

Immersion test basically simulates the underwater zone of the marine environment. It is even more time consuming than the salt spray, because for most coating systems it is even less aggressive due to relatively oxygen depletion. However, it has an outstanding advantage that many electrochemical techniques can be used during immersion to generation more detailed information of coating degradation process. Electrochemical impedance spectroscopy (EIS) is the most popular method used during immersion test, which can reliably monitor the degradation of an organic coating system [152–155] due to its high sensitivity and reliability for a high-impedance system. Due to the ingress of aggressive species or electrolyte, coating resistance decreases and coating capacitance increases [156,157]. After corrosion of the substrate and delamination of the coating, the overall resistance or total impedance and the double-layer capacitance vary further [156,157]. A high corrosion resistant coating system should have slowly decreasing impedance. An equivalent electrical circuit can help understand the degradation behavior of a coating system. With the values of the circuit elements, the degradation degree of a coating system can be quantified [156–162].

To better simulate the marine tidal or splashing zone, cyclic immersion in salt water is also frequently used in the lab, and the degradation of coating systems is usually monitored through intermittent EIS measurements [163–166].

### 6.1.3. AC/DC/AC

Both the immersion and salt spray simulated some marine conditions. They can only speed up the degradation process to a limited extent compared with the natural field marine exposure, and thus have been increasingly criticized for being time consuming [167], particularly when modern organic coatings are becoming more and more corrosion resistant.

AC/DC/AC technique is a combination of DC-driven coating delamination test and immersion EIS measurement to rapidly assess a coating protection performance [168]. It basically cathodically polarizes the organic coating system to speed up the coating blistering or delamination and meanwhile uses EIS to monitor the damage [159,169]. In effect, the AC/DC/AC method is in some senses also a simulation of the coating damage under cathodic protection. In practice, many organic coatings are

used together with cathodic protection. Cathodic hydrogen evolution is the main source of the gas bubbles under the blistered coating. Meanwhile, the alkalization resulting from the cathodic reaction can also facilitate the coating blistering and delamination [170,171].

Since the AC/DC/AC method was first published in 2003 [161], its ability to accelerate coating damage and rapidly evaluate coating performance have been verified by other researchers [160]. For example, differently coated Mg alloys after 1 week (30 cycles) of AC/DC/AC were measured to have corrosion resistance in the same rank as that after 1 month of salt spray, and the EIS spectra in the AC/DC/AC were similar to those during a traditional immersion test [56,151]. These suggested that the AC/DC/AC accelerated the coating degradation, but did not change the coating damage mechanism.

However, the AC/DC/AC can change the corrosion mechanism of the substrate metal. The natural corrosion of a coated metal is a result of anodic dissolution of the metal and cathodic reduction of oxygen, both at the same rate on the metal under the coating. An acceleration of the cathodic process will in the same time depress the substrate anodic dissolution, which differs from the natural substrate corrosion process under the coating.

### 6.1.4. Scratch

Coating defects decisively influence the transfer of corrosive species from the environment through the coating to the coating/substrate interface to trigger the substrate corrosion. It has been reported that under salt spray or immersion condition, coatings with controlled defects can significantly accelerate the substrate corrosion and the coating peeling [172–175]. However, the defects naturally formed during curing are quite random in number, size and distribution, which cannot be easily adjusted to modify the overall corrosion performance of a coating system [67]. To rapidly reveal coating failure mechanism, controllable defects in organic coatings are needed.

A scratch is a macrodefect made on coating to simulate mechanical damage to a coating system during service [176,177]. Such an open defect may be employed to investigate the substrate corrosion after the environmental corrosive species have arrived at the substrate and the coating has peeled off. However, it should be noted that the corrosion damage along the scratch is mainly dependent on the corrosion resistance of the substrate and the adhesion of the coating on the substrate. The test result of a scratched coating does not reflect the permeability or protection performance of the organic coating in theory. Hence, the industrial engineers who still rely on salt spray test to evaluate coating protection performance must treat the results with caution.

A micro mechanical defect can also be made by Atomic Force Microscope (AFM), and its behavior can be monitored by Scanning Kelvin probe technique (SKP) [178]. This microdefect to a great degree simulates the naturally formed coating through pores.

### 6.1.5. UV Irradiation

Since UV can accelerate the oxidation and decomposition of polymers, it can be employed to speed up the degradation of organic coatings [179–181]. It was found that UV irradiation made an organic coating porous [17].

It is more reasonable if various techniques can be used together to accelerate and assess the degradation or damage of an organic coating system. For example, the combination of UV and salt spray may be used to speed up the coating degradation and meanwhile EIS employed to assess and monitor the degradation process [182]. The application of these kinds of multi-techniques will greatly help comprehend the coating degradation behavior and damage mechanism.

### 6.2. Degradation and Damage

In practical service environments, the media can reach the substrate through the coating defects quickly, leading to substrate corrosion and coating blistering or spalling [62,183–185]. Coating degradation can further accelerate the media intake and thus the damage of a coating system.

Coating degradation may be noticed as water-induced swelling and gas accumulation [63,186,187]. The damage can appear as coating chalking, blistering, peeling or spalling, resulting from photo-oxidation [107,108], hydrothermal degradation [109,110], particularly when the coating is subjected to dry–wet alternation [188] and cold–hot cycling [189]. There have been numerous relevant studies under marine conditions [100–102]. In many cases, coating degradation or damage starts from the outmost layer [190]. Hence, the coating surface can to a great extent determine the coating degradation rate [191].

In different marine conditions, a coating may have different degrees of degradation or damage [120]. For example, epoxy resin fully immersed in 5 wt.% NaCl solution ($E_{NaCl}$) was found to have some small "particles" on the surface, while after alternate NaCl solution immersion and UV irradiation ($E_{NaCl+UVA}$) it had a rough surface with a layer of particles. Continuous UV irradiation ($E_{UVA}$) led to many small holes uniformly distributed on the epoxy surface. Alternated exposure in NaCl and air ($E_{NaCl+air}$) resulted in loose product deposits on the surface. After the coating was alternately exposed in water and UV ($E_{water+UVA}$), the damage was relatively mild. If surface roughness is employed to quantify the coating degradation degree, then the degradation processes of the epoxy resin under different exposure conditions can be formulated with a simple equation (see Figure 8) [120]:

$$y = ae^{bt} \tag{1}$$

where $y$ is the average surface roughness of the epoxy resin, $t$ is the weathering time, constant $a$ represents the original surface roughness, and constant $b$ is an accelerating factor associated with the environment condition. According to the equation, the degradation acceleration effect in these different conditions can be arranged in the order of: $E_{NaCl+UVA} > E_{NaCl+air} > E_{UVA} \approx E_{water+UVA} > E_{NaCl}$.

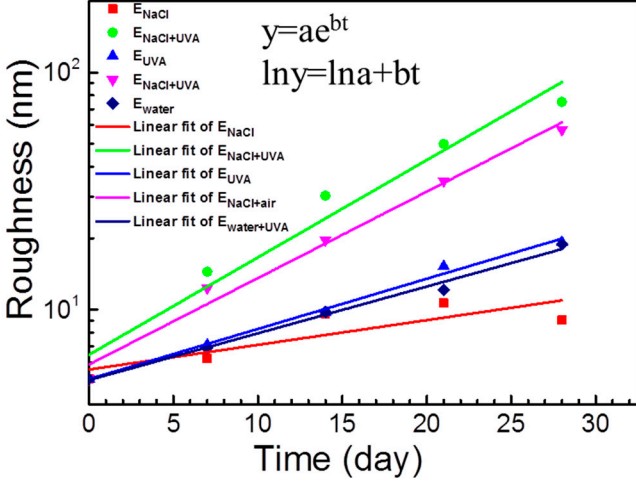

**Figure 8.** Curve fitting of the surface roughness vs. exposure time for epoxy under different weathering conditions, reprinted from [120] (Copyright 2020, with permission from Elsevier).

## 7. Bio-Fouling

Apart from corrosion, bio-fouling is the other annoying problem that the organic coatings have to cope with for the devices and structures in the marine immersion and tidal zones [192–195]. The best way to prevent bio-fouling is to stop the formation or growth of the pre-conditioning film that is essential for bio-fouling species.

### 7.1. Antifouling Coatings

Antifouling coating (AF) is a traditional approach of preventing the attachment of marine biomass or organisms [196]. The incorporated biocide additive is a critical constituent in such coatings, which inhibits the micro- and macro-fouling behavior on the coating surface [197]. $Cu_2O/CuO$ and $ZnO$

are traditional biocides and still being used somewhere [198]. They have been made into nanoparticles to further improve their anti-biofouling efficiency [199,200]. Unfortunately, most of the biocides, like $Cu_2O/CuO$, ZnO, zinc pyrithione and copper pyrithione, tributyltin, diuron, and irgarol, are not environmentally friendly [198–202]. Due to increasing environmental concerns, many previously used biocides have been, or are going to be banned [203,204], which is stimulating the development of new biocide-releasing coatings. Coating developers have to seek for new environmentally friendly additives or biocides, such as antibiotics, quaternary ammonium ions, and some inorganic ions, compounds or particles [205–209]. To avoid possible pollution to the environment, it is import that the additives must be highly effective and strictly selective to fouling. A very small addition is already sufficient.

Another eco-friendly approach to antifouling is the fouling-releasing coating [210–213]. For example, silicone-based polymers can be developed into a fouling-releasing coating to release attached bio-foulings easily from the coating surface [214]. The composite poly(dimethysiloxane) coating with $SiO_2$-ZnO nanoparticles has been reported to have self-cleaning and antifouling behaviors [215]. However, the coatings are costly and not strong enough in practice [216].

Currently, antifouling coating development has also been focused on self-polishing [217,218], environmentally friendly biocide-releasing antifouling [200,219–221], non-fouling [19,222,223] and self-healing [16,224] functions.

### 7.2. Self-Polishing and Biocides-Releasing Coatings

A self-polishing coating generally consists of acrylic and meth-acrylic copolymers that can easily be hydrolyzed in seawater [225,226]. The biocide embedded in the coating is gradually exposed and released to kill the biofoulants. Meanwhile, the surface layer of the coating running out biocide and already attached by biofoulants is polished/removed off to expose fresh coating surface containing sufficient biocide.

Although, nanoparticle inorganic biocides have higher antifouling efficiency and can reduce the required amounts, they are toxic to non-targeted organisms [227,228]. Some nontoxic organic biocides have been proposed [229]. They are expected to accelerate the degradation of protein molecules of biofoulants, reduce the adsorption Gibbs free energy of the degraded peptide fragments, and consequently prevent or slow down the formation of the pre-conditioning film to inhibit the growth of fouling organisms. For example, serine protease can be encapsulated and subtilized in a sol-gel coating, and applied on a stainless steel surface (see Figure 9), to retain the enzyme activity against the growth of biofilm for 9 months [230].

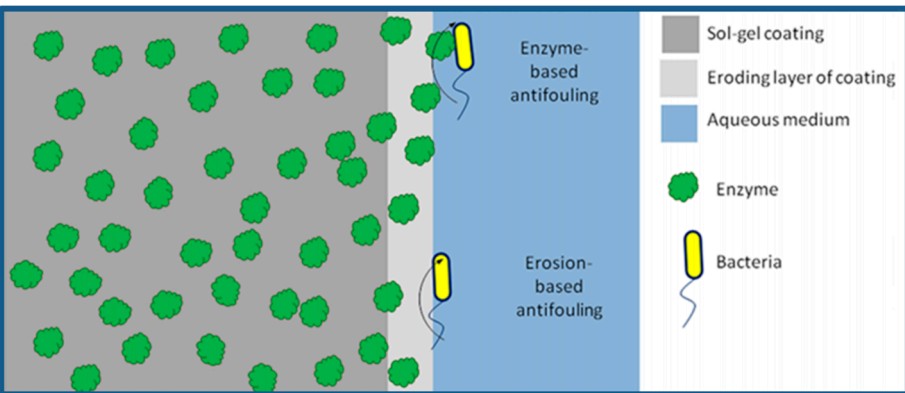

**Figure 9.** Schematic illustration of the antifouling mechanism of a serine protease encapsulated sol-gel coating, reprinted from [230] (Copyright 2020, with permission from American Chemical Society).

### 7.3. Non-Fouling Coatings

All the self-polishing coating and biocides will run out eventually. A permanent solution of biofouling may exist in coating surface functionalization [231–237]. Therefore, the research in this

direction has attracted many efforts and resources. For example, hydrophobic coatings (with a water contact angle greater than 90 degree), such as silicone-based and fluorine-based coatings, have low surface energy, low elastic modulus and high surface roughness. The lower surface energy implies less active interaction with live species. When a surface is rough enough, the accumulated biofoulants can be easily removed in a hydrodynamic fluid [238]. With low elastic modulus and critical surface free energy, the coating surface will not be sticky to attached species [239]. Duong et al. [240] found that a synthesized poly(dimethylsiloxane) (PDMS) coating was hydrophobic in artificial seawater, having good anti-adhesion of bacteria and antifouling activity. Although high hydrophobic and low elastic modulus coatings are environment friendly antifouling, they are not wear- or scratch-resistant in static water, and thus cannot find wide practical applications.

A hydrophilic polymer "brush" can prevent adhesion of proteins due to the entropic repulsion, and thus it can avoid the formation of a biofilm for biofouling attachment [241]. Poly ethylene glycol (PEG) is one of the most commonly used hydrophilic polymers. PEG-based coatings can be obtained from self-assembled monolayers (SAMs) through atom transfer radical polymerization (ATRP) to graft the PEG brushes on a substrate surface. Therefore, the resistance of the surface against protein adhesion increases with the density of the grafted PEG and its chain length [242]. Benjamin et al. [243] fabricated PEG layers step by step on a silicon surface to significantly improve the stability and density of PEG in order to optimize the antifouling behavior (see Figure 10). It was recently further reported that a loop surface had better anti-protein performance than a brush-bearing surface [244].

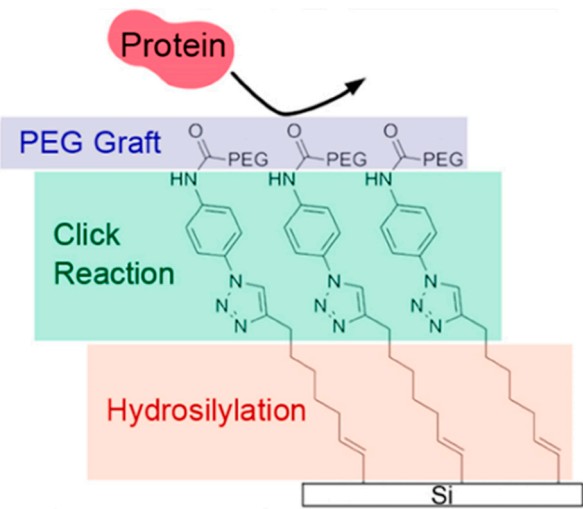

**Figure 10.** Schematic illustration of the PEG layers on silicon, reprinted from [243] (Copyright 2020, with permission from American Chemical Society).

The biomimetic attachment of PEG to various substrates interests some antifouling coating developers. Trimeric catecholates have been synthesized and immobilized on TiO$_2$ and stainless steel, and PEG conjugated to a trimeric catecholate surface to prevent the adsorption of human blood and bacteria [245]. Although most hydrophilic brush polymer systems have satisfactory antifouling performance, they have some obvious drawbacks, such as limited mechanical robustness, high sensitivity toward hydrolysis and oxidation, and selected effectiveness for only a few bio-fouling species.

In recent years, several zwitterionic materials including sulfobetaine (SB), phosphorylcholine (PC), and carboxybetaine (CB), with both anionic and cationic groups were investigated for their resistance to protein adsorption [246–248], and found that their strong hydration and charge balance could reduce the entropy gain and the enthalpy loss against protein adsorption [249]. Two factors should be considered for a zwitterionic coating. Firstly, the protein resistance of a zwitterionic brush is also affected by the grafting density and chain length like hydrophilic brush. Secondly, a hydrophilic

zwitterionic polymer coating tends to be dissolved in environmental water to gradually expose the underneath fresh surface. When a zwitterionic macro-crosslinker is applied on a polyurethane substrate, it will be much thicker and more durable than a common polymer brush coating, maintaining its hydrophilicity for more than two weeks in static water and retaining its resistance to the adsorption of proteins and bacteria for at least one week in flowing condition [250,251]. Zhao et al. [252] grafted zwitterionic sulfobetaine vinylimidazole (pSBVI)-based polymer brushes to a silicon substrate through electrochemical surface-initiated atomic-transfer radical polymerization (e-SIATRP) to reduce the adhesion of marine alga nannochloropsis maritima (see Figure 11). The e-SIATRP technology appeared to be more effective than a traditional ATRP as the reaction solution is recyclable.

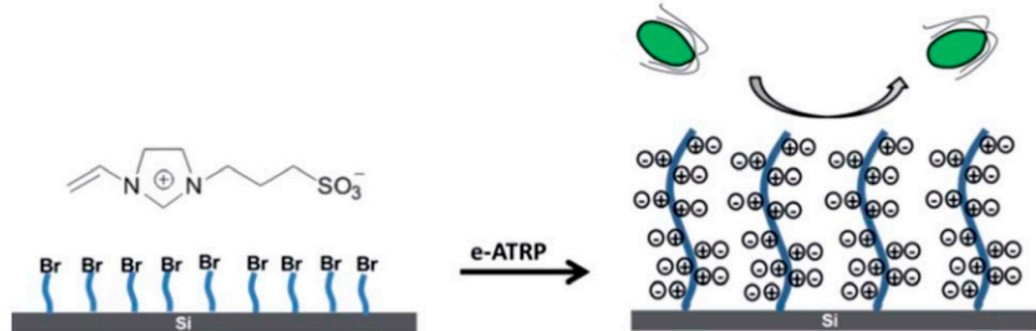

**Figure 11.** Grafted zwitterionic polymer for antifouling, reprinted from [252] (Copyright 2020, with permission from Royal Society of Chemistry).

Poly-cationic antibacterial materials obtained from quarternization of tertiary amine groups with alkyl halides have also an antifouling behavior, because the negatively charged cell membranes of bacteria could be damaged by the highly positively charged polymer chains [253]. They can be incorporated into polymers, films, dendrimers and particles on various substrates to make the surfaces antibacterial. Many researchers are trying to enhance the polycationic antibacterial activity [254,255]. For example, Yuan et al. [256] synthesized hydroxyl-rich cationic derivatives via the ring-opening reaction of a star-like poly (glycidyl mehacrylate) (s-PGMA) to a form antibacterial and antifouling coating (see in Figure 12).

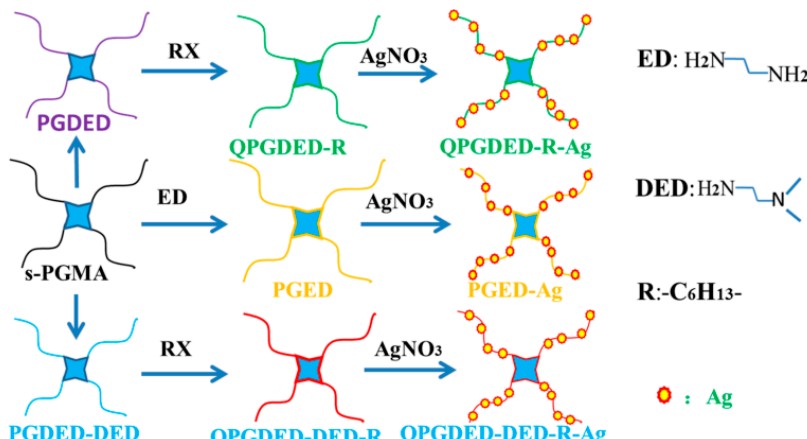

**Figure 12.** Schematic illustration of the formation processes of different hydroxyl-rich antibacterial polymers via the ring opening reaction of s-PGMA, adapted from [256] (RX: Haloalkane, PGMA: poly (glycidyl mehacrylate), PGDED and PGDED-DED: the products after DED and DED/ED being introduced into s-PGMA, QPGDED-R, PGED and QPGDED-DED-R: the products modified by RX and ED, QPGDED-R-Ag, GED-Ag, QPGDED-DED-R-Ag: the products modified by AgNO$_3$).

Either a hydrophilic or hydrophobic surface is antifouling to only some species [257]. If the surface is made amphiphilic with a combined hydrophilic and hydrophobic effect, the antifouling resistance may be further improved [258–260]. Amphiphilic polymers general contain fluorinated and polyethylene glycol (PEG) or polyethylene oxide (PEO) based groups with an optimized ratio of hydrophobic and hydrophilic segments. The amphiphilic surfaces may be fabricated by a layer-by-layer self-assembly (LbL) method, which have a poly-anion grafted to amphiphilic fluoroalkyl polyethylene glycol (fPEG) side chains, and can effectively prevent the adhesion of marine bacterium Pseudomonas (NCIMB 2021) [261]. Chen et al. [262] synthesized a ternary copolymer composed of N-(4-hydroxy-3-methoxybenzyl)acrylamide (HMBA), poly(ethylene glycol) methyl ether methacrylate (PEGMA), and 2-hydroxyethyl acrylate (HEA) via facile radical polymerization, which was further cross-linked with PAPI to form polyurethane (PU) coatings (see Figure 13). The prepared copolymer coatings exhibited heterogeneity, amphiphilicity and excellent fouling resistance to protein Bovine serum albumin (BSA).

**Figure 13.** Synthesis route of P (H-P-A) and PU films, reprinted from [262] (Copyright 2020, with permission from American Chemical Society).

## 7.4. Self-Healing Antifouling Coatings

Even a non-fouling coating can lose its antifouling efficiency in the mechanically damaged areas. It will be ideal to have a coating that can resume its originally designed functions in the mechanically damaged local area. Inspired by some living organisms that can survive, revive or recover after being seriously hurt or partially damaged, researchers are now trying to develop some self-healing materials for coatings [263–269]. Chen et al. [16] developed an underwater superoleophobic and self-repairing anti-biofouling coating for the first time through self-assembly of the hydrophilic polymeric-chain-modified hierarchical microgel spheres (MHMSs) (see Figure 14). Xue et al. [224] also fabricated a robust self-healing superhydrophobic poly (ethylene terephthalate) (PET) fabric via a convenient solution-dipping method using widely available polydimethylsiloxane (PDMS) and octadecylamine (ODA), which exhibited remarkable durability to abrasion, washing in different pH solutions, and could resume its superhydrophobicity even after air-plasma treatment.

Currently, most of the environmentally friendly antifouling coatings are more successful in the lab than in field. There is still a long way to cost-effective long service life antifouling coatings for practical marine applications.

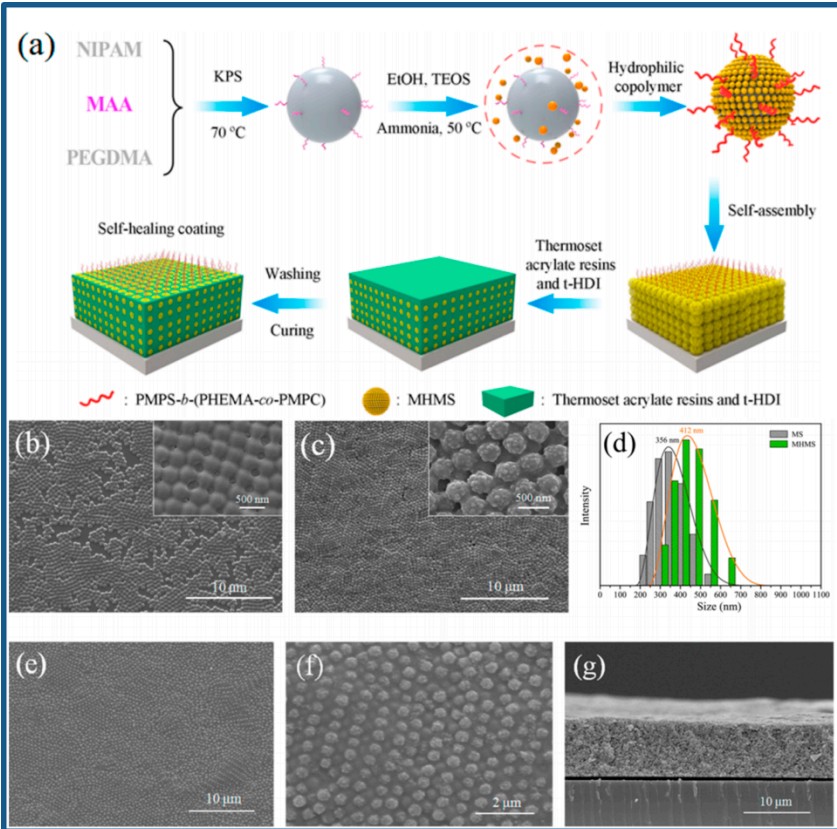

**Figure 14.** Schematic illustration of the preparation of a MHMS-based coating (**a**), SEM images of the microgel spheres (MS) (**b**) and MHMS (**c**), size distribution of the MS and MHMS (**d**), SEM image of the MHMS-based coating (**e**), the magnified SEM image (**f**) and cross-sectional SEM image (**g**) of the MHMS-based coating, reprinted from [16] (Copyright 2020, with permission from Springer Nature).

### 7.5. Evaluation of Antifouling Coatings

Coating characterization is critical to the design and development of organic coatings [270], which can help optimize coating chemical composition and microstructure [271], surface morphology [272] or surface energy [273]. Generally, the molecular structures of organic coatings can be detected by attenuated total reflectance Fourier transform infrared spectroscopy (ATR-FTIR) [137,139], X-ray Photoelectron Spectroscopy (XPS) [274] and proton nuclear magnetic resonance spectroscopy ([1]H-NMR) [275]. The inorganic additives in antifouling organic coatings, such as $Cu_2O$ or ZnO, can be detected by X-Ray Powder Diffraction (XRD), thermogravimetric analyzer (TGA) and atomic absorption spectroscopy [198,256]. The coating surface morphology can be analyzed by atomic force microscopy (AFM) [141,142], scanning electron microscopy (SEM) [276], laser scanning confocal microscopy (LSCM) [277] and fluorescent microscopy [278], and the surface energy can usually be estimated from the contact angle of water or methylene iodide [279]. The coating strength and bonding force can be determined by tensile test [280], adhesion test [281] and dynamic mechanical analysis (DMA) [180,282].

Antifouling performance is one of the most critical properties determining the success of a coating for marine applications. The antifouling performance evaluation methods include field tests and laboratory-based antifouling assays [283]. The field tests are more reliable, while laboratory methods are time-saving. A field assay can be either dynamic or static. In the static immersion, the designed antifouling specimens are exposed 1 m deep in seawater. During dynamic immersion, the specimen panels are attached to a moving float, exposing to the seawater parallel to the water flowing direction. In the laboratory, the traditional atomic force microscopy (AFM) [141,142], spinning disk [284], hydrodynamic shear force [285] can be applied to coating surfaces to quantify the adhesion of biological species.

In addition to the above conventional coating characterizations and performance evaluations, many new techniques had been developed to assess the antifouling behavior. Alles and Rosenhahn [286] established a microfluidic assay to quantitatively characterize diatom adhesion on different antifouling surfaces. In the method, micro-channels were formed by polydimethylsiloxane (PDMS) gaskets squeezed between a glass-lid and the sample surface. Thus, the shear stress on the channel walls could be calculated [287]. This sensitive method only requires a small volume of diatom suspension for test. George et al. [288] theoretically simulated fouling-release behavior on polymer surfaces, and demonstrated that the surface fouling-release was determined by the surface affinity to biofoulants and water. The observed wetting and adhesion phenomena could be related to the nanoscale surface features. A computational fluid dynamics (CFD) model was also proposed to predict the friction resistance of antifouling coatings based on surface roughness measurements [289,290].

The above methods are mainly for lab studies. Their reliability needs verification in field. It is highly desired that new methods as informative as the above lab techniques can be developed for reliable field tests.

## 8. Concluding Remarks

From a corrosion protection point of view, the defects ranging from nano-level broken molecular bonds to micro-level gas bubbles and shrunk pores, up to even millimeter-level cracks are the decisive factor for an organic coating. The degradation or damage of the organic coating is in nature an exacerbation process of the defective effect, which eases the travel of corrosive species through the coating from the environment to the substrate metal.

Since there are inevitably many short-cut defects in an organic coating and the accumulation of corrosion products on the substrate can facilitate the coating degradation and damage, the corrosion resistance of the substrate can affect the coating protectiveness. The coating protection performance can be improved by using a more corrosion resistant substrate or by pre-treating the substrate to enhance its passivity before coating or even simply by reducing the substrate corrosion damage during coating.

With the coating defects, the corrosive media can reach the substrate eventually. Thus environmental factors can sooner or later influence the substrate corrosion through the defects. Meanwhile, some environmental factors and processes can also make the coating more defective and accelerate the coating degradation. Particularly when the coating degradation is facilitated by multi environmental factors, there may be a synergistic acceleration effect on the coating decay and failure. The worst case is that the multi environmental factors simultaneously accelerate the substrate corrosion and coating degradation, which will lead to rapid coating delamination or spallation.

To predict the corrosion damage of a coating system, the most useful information is the defect size, number and distribution in the coating. However, direct determination of the nano- and microdefects in an organic coating on a metal is difficult. Even worse, a quality coating system usually degrades or corrodes slowly. In order to obtain enough results to predict the coating system's long-term performance, the degradation and damage should be sped up. Unfortunately, it is highly risky to predict the long-term coating protection performance based on limited short-term test data.

Antifouling research has been focused by many coating researchers and engineers for decades. Non-fouling coating and self-repairing coatings are currently and will continue to be hot research topics. Based on the innovative ideas that have been proposed and tested, it is more important to develop some long-term cost-effective antifouling coatings in real marine environments.

Additionally, the rapid evaluation methods that can truly and rapidly characterize and assess the bio-fouling behavior of antifouling coatings in real marine environments are urgently needed.

## 9. Future Perspective

Current rapid development of the marine industry demands for more corrosion resistant organic coatings with more effective antifouling performance for a wider variety of marine devices and structures. There is no doubt that more research resources will be invested in new coating development

in future. It is likely that coating researchers and engineers will make every endeavor to further improve the coating technology in the following areas.

### 9.1. Coating on New Marine Material

It is believed that more new materials will be used in marine environments. Due to the differences in physical and chemical properties, the coating systems required for them may be different from those for the traditional marine materials. For example, light weight Mg alloys [160,291–298] may be increasingly used in aircrafts and helicopters on marine gas/oil platforms, warships and aircraft carriers. They need very robust coatings in the marine atmosphere, as they are much more active than the traditional substrate metals and behave differently in electrochemical experiments [158], which can significantly influence the coating formation and protection performance. By utilizing the special surface alkalization effect of Mg, a new coating technique, Electroless E-coating process, has been developed for Mg alloys [86,158,162,299–306], in which surface wetness is critical to the coating performance [89,158,299,300]. Obviously, different substrates will have different electrochemical characteristics, which may lead to different coating formation and protection behaviors.

### 9.2. Environmentally Friendly Coating

Some toxic or polluting additives and compounds may be released into environment from coating materials in the coating system manufacturing, applying or using stage. The organic coating industry has been fighting for "green" products for decades. The importance of this environment hazard issue will be never overstressed in future.

Tremendous efforts have been made to advance environmentally friendly organic coating technology in the traditional corrosion prevention field [307]. Unfortunately, the antifouling requirement strictly limits the options of coating materials and manufacturing procedures. Therefore, in future the antifouling organic coating technique and system will continuously be a challenging research topic.

There are already some promising antifouling coating materials recently reported, such as exfoliated GO-$Al_2O_3$ hybrid sheet and $MnO_2$-silicone hybrid polydimethylsiloxane film [308,309], silver nanoparticle [206,310–315], silver-doped graphene oxide nanosheet-chitosan nanoparticle [316]. Natural raw materials appear to be quite attractive to coating researchers [202,317,318]. The green materials are more likely to make an antifouling coating environmentally friendly. Along this approach, some cost-effective and long-term effective antifouling coatings may be developed for marine applications in future.

### 9.3. Rapid Reliable Coating Valuation

When the degradation of a coating system is accelerated artificially, the natural failure mechanism is likely changed. For example, temperature elevation may lead to polymer crystallization, scratching will completely eliminate the local barrier effect of a coating, and ultra-sounding can mechanically crack the coating or peel off the coating, all of which completely change the transfer mechanism of corrosive media in the coating. A system tested by means of these rapid methods will degrade and fail differently from its normal decay behavior, and the tested results will be meaningless. It is critical that an accelerated test method can significantly speed up the damage of a coating system, but not change its normal degradation mechanism. Obviously, this is difficult. Hopefully, it can be achieved after all the detailed coating degradation and damage mechanisms are comprehensively understood. For example, it is well-known that in principle the anodic polarization current density of the substrate metal is always equal to the cathodic polarization current density for a coating system in a corrosive environment. After a long-term natural exposure, the total electric charge from the anodic polarization should be equal to that from the cathodic polarization. More the charge produced means worse corrosion damage to the substrate, and a larger amount of oxide, hydroxide or hydrogen accumulated to peel off or crack the coating. Therefore, to reasonably accelerate the corrosion damage of the coating, the DC polarization in the AC/DC/AC test should be modified to equally enhance the anodic

and cathodic processes, rather than accelerating the cathodic reaction but decelerating the anodic dissolution. With this understanding, the AC/DC/AC method may be optimized by a combination of cathodic and anodic polarization in the DC step.

### 9.4. Service Life Prediction

Corrosion behavior is always difficult to predict. Service life is the most challenging topic in corrosion science and protection engineering, particularly for an organic coating system with many complicated various and varying influencing factors.

Generally speaking, for corrosive species from the environment to corrode the substrate metal under an organic coating, there are 3 stages (see Figure 15): (1) $t_s$-ingress into the coating, (2) $t_c$-diffusion through the coating, and (3) $t_i$-interaction with the substrate and initiation of corrosion. After that, coating will start blistering, peeling and spalling, and gradually lose its protectiveness, although sometimes the blistering, peeling or spalling process is slow and may not be easily detected. Therefore, the service life of the coating system is determined by these stages [156,319]:

$$t = t_s + t_c + t_i \tag{2}$$

where $t_s$ is dependent on the coating surface state, $t_c$ is determined by the coating porosity and thickness, and $t_i$ is controlled by the substrate state. The $t_s$ may be significantly shortened in the lab by increasing the concentrations the corrosive species, which does not change the ingress mechanism. The $t_i$ may be sped up by a specially designed electrochemical process. The most difficult step is how to speed up the $t_c$ stage, which is a diffusion process in the coating body. Limited by the surface layer of the coating, the diffusion cannot be sped up significantly by increasing the concentrations of those species from the environment, and the organic coating cannot be heated up too much to evidently accelerate the diffusion either. If the species transfer in the coating body is assisted by an electric, magnetic or other field, the diffusion process may be dominated by migration, and the transfer mechanism will be changed.

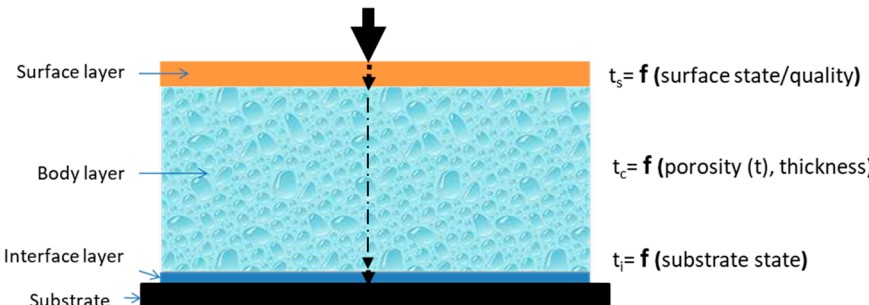

**Figure 15.** Schematic illustration of the ingress of corrosive species into coating through the surface layer, the diffusion of corrosive species in the coating body and the interaction of corrosive species with the substrate through the interface layer.

Unfortunately, almost all the organic coatings used in marine environments are thick with low permeability. Corrosive species cannot easily penetrate the coating. Without changing the slow diffusion mechanism of the corrosive species in the coating and the corrosion mechanism of the substrate metal to dramatically shorten these two stages, it will be extremely difficult to monitor the degradation and damage of the coating system. Without sufficient degradation and damage information, there is no way to predict the coating system's service life. Hopefully, in future work, this hurdle for coating systems in marine applications will be targeted by more coating researchers.

**Funding:** National Key Research and Development Program of China (Grant No. 2017YFB0702100), the National Natural Science Foundation of China (key project Grant No.51731008 and general project Grant No.51671163),

National Environmental Corrosion Platform of China, and the National Key Research, Science and Technology Planning Project of Fujian Province (2018H6017).

**Acknowledgments:** The authors also appreciate the assistance from Dajiang Zheng, Yuqing Xu, and Naiyuan Fang in some lab experiments.

**Conflicts of Interest:** The authors declare no conflict of interest. The funders had no role in the design of the study; in the collection, analyses, or interpretation of data; in the writing of the manuscript, or in the decision to publish the results.

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
