# Peer review of "Modification, Degradation and Evaluation of a Few Organic Coatings for Some Marine Applications"

_cmd, doi:10.3390/cmd1030019_

Round 1

Reviewer 1 Report

The review is very interesting and gives an important update in the field of marine coatings. I feel that it could be improved with a first short section in which the main organic coatings used for corrosion protection are mentioned, as well as the main substrates; this would be an useful introduction also for the non-specialized reader. I like the choice of giving advice at the end of each section, it is useful and practical.

After reading the entire manuscript, I am a bit puzzled by the title because reading “Modification and Evaluation of Organic Coatings for Marine Applications” I was expecting a description of techniques to chemically or physically modify existing organic coatings and to test their activity. Maybe the authors should think and consider something like “Degradation of Organic Coatings for Marine Applications and evaluation of their efficiency/performance”. The title does not refer neither to the important section about the antifouling topic.

As the authors say that they want to present “recent” advances, they should say in the introduction in which period this bibliographic research spans.

Abstract: first sentence, if “marine” is used as adjective, the sentence has no meaning; if it is used as noun, the sentence should be rewritten because it is not clear, and it is the first thing we read.

I would suggest modifying keywords, and not writing “marine” and “defect” alone.

Introduction: lines 28 et 29, “ For example, an organic coating for marine applications is usually thicker”, thicker than what? Please give an order of magnitude or finish the comparison.

Minor corrections

Line 180, RH is repeated, eliminate one.

Line 197, line 404, and so on write et al. in italics.

Line 257, interesting instead of interested

Line 449, correct “Many research researchers”

Figure 12 contains too many acronyms that are not written so that the meaning is not clear unless you have read the paper; it could be eliminated or the meaning of all acronyms should be clearly explained.

Reference 283, change police

Reference 209, change the bold characters for the journal name

Reviewer 2 Report

Dear authors, 

The issues described on this paper are not novel. There are several books and papers published on this specific area. However, at the end of the papers the authors include some new technologies that were recently applied to design and evaluate coatings for marine applications. 

Abstract

The abstract should be more informative, what are the major conclusions about organic coatings? What did you find out with this research?

Introduction

Please the introduction is too short, the authors should include more info. I would like to know why this review is so important.

What are the major substrates, organic coatings, and what about their defects? That should be stated somehow in the introduction.

How do substrates interact with organic coatings? Why organic coatings are the most used ones?

L55-57 What are the corrosive species?

Figure 3. Capitalize the “schematic” on the Title.

Review

Please reorder the topics

You haven’t mentioned any defect yet and you are already talking about defect modification.  I would suggest change the order to:

2) Substrate Effect; 3) Environmental Influence; 4) Defect Modification; 5) Bio-Fouling; 6) Evaluation

Reviewer 3 Report

See the attachment

Reviewer 4 Report

With an appealing title, this review does not bring what it promises. No one working in the field of marine paints will find it worth reading. This is because in 284 references only a few of them actually address marine applications, marine coatings or even polymers typically used in marine coatings. On the other hand, count the number of papers with magnesium (hardly used in marine applications). Most of the advances reported here might be interesting for organic coatings in general, not so much for the particular niche of marine applications. This work reviews paints in general not marine coatings.

With a title like this it is important (mandatory) to focus on the paints (polymers) that are actually used in marine conditions. And make note that there are primers and topcoats. A complete paint scheme is composed by at least one layer of each one.

We can divide the review in two parts, one focusing corrosion, and another focusing biofouling. The biofouling part seems to be better accomplished, however, also here the link to marine coatings is weak.

This paper benefits by starting with a very short overview of:

  1. a) how paints protect metal from corrosion
  2. b) how paints degrade
  3. c) current paints used today in marine applications
  4. d) the main problems associated with current coatings in marine applications

Then it becomes easier to identify the pertinence of the papers being reviewed.

And, above all, this will prevent a problem in the current version of the manuscript, which is the presentation of a) and b) in different parts of the text, sometimes as if it is novel knowledge and not facts known for decades.

These are the main reasons for recommending the rejection of the current version of the manuscript. A substantial review is necessary to make it worth reading.

A finer analysis to the manuscript is now presented.

The first references appear to support general concepts, but it is not clear the criteria to choose them among so many possible papers. For example, reference 2 is given to support the idea that an organic coating is an effective barrier to corrosion media. Instead of books, review papers or research papers directly studying the barrier properties of paints (and there are plenty) why this paper?

Reference 3 is not about organic coatings.

Reference 17 is for biomedical applications.

References 18-34 are supposed to be there to show the tremendous research efforts to improve the performance of organic coatings in marine environments. Check them!

Section 2.

Figure 1. The pictures are from top surfaces or cross sections? They show holes, craters, defects or pores? If pores, they cross the entire thickness of the paint?

More important, why show pictures of powder coatings or e-coatings, when their use in the marine applications is rare?

Page 2, line 50-51. Give reference to support this sentence.

Page 2, line 63-64. About the use of temperature, pressure or airflow to solve the problem of defects across the paint film: the simplest way is to use various layers of coatings. In marine applications thick coatings also decrease de risk of appearance of defects crossing the total paint scheme. And if defects can reach the surface, they must be so large that difficulty the above solutions will work. These are interesting solutions for the paint area in general, not that much applicable to the marine applications.

Figures 2 and 3 give results of alkyd paints, which are too weak to be used in marine environments. Again, this is good for a review on paints but not for for marine paints.

Section 3. Substrate effect.

This is old knowledge, common sense nowadays. It should not be presented as if it is something new.

Figure 4, lines 116-124. That is why there are strict temperature and humidity conditions that applicators must obey to avoid these and other problems.

Lines 128-142, including Figure 5. What´s the relevance for marine coatings?

Reference 71 appears supposedly for the reader to learn more about the “tremendous research and engineering efforts that have been made to develop more cost-effective substrate pretreatments for coatings”. Shouldn’t a book, handbook, review paper(s) be better references than a single paper about “surface preparation of machine parts by ultrasonic impact treatment before coating”?

Page 6, lines 176-192 are not that relevant for this review.

Note that in conditions of high RH marine paints are not applied, if the best practices (and product datasheets) are followed.

Relevant points to analyze with respect to water is additives for water repellence; behaviour of paints in high RH versus under complete immersion; the variations in osmotic pressure and coating properties when a ship/boat passes from fresh water to sea water and vice-versa.

Also differences in the environment aggressiveness (and coating behaviour and durability) moving from submersed, splash and spray zones.

  1. Evaluation.

References 100-102. Too poor. 3 papers? There are books on the subject. Annual meetings on the subject with special issues published on a regular basis. Why these 3 papers?

In lines 231-236 you refer electrochemical methods but then forget to name them. Non-electrochemical ones are not forgotten. What could be interesting here is referring the advantages and disadvantages of all these techniques. What do they provide? And mention those that can be used in service and those that can be used only in the laboratory.

Page 8, lines 249-357 (Salt Spray).

Again, emphasis on magnesium substrates and nothing with steel which is the main substrate in marine applications. Examples with steel and “naval steel” as substrates can be found.

Check the references again.

Just two examples: First reference, the name is Thierry, not Thieny.

References 43 and 138 are the same.

Round 2

Reviewer 2 Report

The authors accepted the suggestions and the paper significantly changed. 

Reviewer 4 Report

Taking into account the authors’ response about the use of Mg alloys in some niche marine applications (helicopters in warships) the title should directly refer to it.

Maybe like this (and not pretending to address the general and wide area of marine coatings) the manuscript has a chance. Still, not many people in the field of marine coating will profit in reading it. The acceptance should be an editorial decision.